# Outcome of Completion Surgery after Endoscopic Submucosal Dissection in Early-Stage Colorectal Cancer Patients

**DOI:** 10.3390/cancers15184490

**Published:** 2023-09-09

**Authors:** Nik Dekkers, Hao Dang, Katinka Vork, Alexandra M. J. Langers, Jolein van der Kraan, Marinke Westerterp, Koen C. M. J. Peeters, Fabian A. Holman, Arjun D. Koch, Wilmar de Graaf, Paul Didden, Leon M. G. Moons, Pascal G. Doornebosch, James C. H. Hardwick, Jurjen J. Boonstra

**Affiliations:** 1Department of Gastroenterology and Hepatology, Leiden University Medical Center, 2333 ZA Leiden, The Netherlandsj.j.boonstra@lumc.nl (J.J.B.); 2Department of Surgery, Haaglanden Medical Center, 2512 VA The Hague, The Netherlands; 3Department of Surgery, Leiden University Medical Center, 2333 ZA Leiden, The Netherlands; 4Department of Gastroenterology and Hepatology, Erasmus MC Cancer Institute, University Medical Center, 3015 GD Rotterdam, The Netherlands; 5Department of Gastroenterology and Hepatology, University Medical Center Utrecht, 3584 CX Utrecht, The Netherlands; 6Department of Surgery, IJsselland Hospital, 2906 ZC Capelle aan den IJssel, The Netherlands

**Keywords:** colorectal cancer, T1CRC, endoscopic submucosal dissection, completion surgery, morbidity, nationwide database, total mesorectal excision

## Abstract

**Simple Summary:**

Instead of extensive conventional surgical resection, early-stage colorectal cancers are now often primarily treated using specialized local resection techniques, such as the endoscopic submucosal dissection (ESD). Sometimes after ESD a regular surgical resection is still needed. However, the impact of ESD on this surgery has not been well studied yet. This study aimed to investigate if ESD affected the safety and outcome of completion surgery. Outcomes of two groups of patients were compared: one consisting of patients who only had an upfront surgical resection and another consisting of patients who had an ESD followed by a surgical resection. Results showed that safety and outcome of surgery were similar in both groups. This means that ESD does not significantly increase negative outcomes of surgery. This knowledge empowers doctors to perform ESD as a first treatment option for early-stage colorectal cancers.

**Abstract:**

T1 colorectal cancers (T1CRC) are increasingly being treated by endoscopic submucosal dissection (ESD). After ESD of a T1CRC, completion surgery is indicated in a subgroup of patients. Currently, the influence of ESD on surgical morbidity and mortality is unknown. The aim of this study was to compare 90-day morbidity and mortality of completion surgery after ESD to primary surgery. The completion surgery group consisted of suspected T1CRC patients from a multicenter prospective ESD database (2014–2020). The primary surgery group consisted of pT1CRC patients from a nationwide surgical registry (2017–2019). Patients with rectal or sigmoidal cancers were selected. Patients receiving neoadjuvant therapy were excluded. Propensity score adjustment was used to correct for confounders. In total, 411 patients were included: 54 in the completion surgery group (39 pT1, 15 pT2) and 357 in the primary surgery group with pT1CRC. Adverse event rate was 24.1% after completion surgery and 21.3% after primary surgery. After completion surgery 90-day mortality did not occur, though one patient died in the primary surgery group. After propensity score adjustment, lymph node yield did not differ significantly between the groups. Among other morbidity-related outcomes, stoma rate (OR 1.298 95%-CI 0.587-2.872, *p* = 0.519) and adverse event rate (OR 1.162; 95%-CI 0.570-2.370, *p* = 0.679) also did not differ significantly. A subgroup analysis was performed in patients undergoing rectal surgery. In this subgroup (37 completion and 136 primary surgery), these morbidity outcomes also did not differ significantly. In conclusion, this study suggests that ESD does not compromise morbidity or 90-day mortality of completion surgery.

## 1. Introduction

A growing number of early-stage colorectal cancer (CRC) patients are primarily treated with a local resection instead of major surgery [1]. Endoscopic submucosal dissection (ESD) is an increasingly popular local resection technique that can be used to resect suspected T1CRCs *en bloc*, regardless of their size. These ESDs can be considered as definitive treatment for a portion of these T1CRCs [2,3]. However, if either resection margins are positive, indicating a possibility of an incomplete resection, or if high-risk features for lymph node metastasis (LNM) are present, current guidelines recommend completion surgery [4,5]. These high-risk features include poor differentiation, deep submucosal invasion, high-grade tumor budding and lymphovascular invasion [5]. In current practice, there is an indication for completion surgery in more than half of the T1CRC patients after ESD [6].

Whether a prior ESD affects the outcome of possible completion surgery has been a topic of discussion. Multiple studies have shown the long-term safety of ESDs [7,8,9,10,11], but the influence of ESD on surgical morbidity remains unclear. Morbidity rates of completion surgery following local resections have mostly been studied for prior local surgical resections, such as transanal endoscopic microsurgery (TEM) or transanal minimally invasive surgery (TAMIS). A recent meta-analysis on this subject illustrated that prior local surgical resections increase the complexity of completion surgery, leading to increased procedure times and an increased adverse event rate compared to primary oncological resections [12]. It was hypothesized that the preceding local surgical resections caused inflammatory changes that could lead to scarring and fibrotic changes surrounding the previous resection site, resulting in adhesions and challenges for dissection of the correct anatomic planes and performing anastomosis [13]. This raises the question of whether the same applies to a preceding ESD, despite the more superficial dissection plane of the submucosa.

The aim of this study is to compare the morbidity of completion surgery after ESD to primary surgery in a Western population of suspected T1CRC patients, using data from a nationwide database and propensity score adjustment to correct for baseline differences between both groups.

## 2. Materials and Methods

### 2.1. Population

A retrospective cohort study was performed. Approval for this study was obtained from the institutional review board (IRB) of the Leiden University Medical Center (reference G18.097) and the Dutch ColoRectal Audit (DCRA; reference DCRA202015). The need for informed consent was waived by both IRBs. 

#### 2.1.1. Completion Surgery Group

Patients in the completion surgery (CS) group were selected from a prospective database of consecutive ESD procedures from three tertiary hospitals in the Netherlands: Leiden University Medical Center (LUMC), Erasmus Medical Center (EMC) and University Medical Center Utrecht (UMCU) between 2014 and 2020. Patients who underwent completion surgery after ESD for suspected T1CRC, located in the rectum or sigmoid, were selected. Exclusion criteria were neoadjuvant therapy and missing data on ≥5 outcome variables.

#### 2.1.2. Primary Surgery Group

The primary surgery (PS) group consisted of patients from a nationwide database for surgical data, the DCRA database (January 2017 and December 2019). More information regarding the methodology, quality checks and external validation of this nationwide registration has been described previously [14,15]. Patients who underwent primary oncological resection for pT1CRC, located in the rectum or sigmoid, were selected. Exclusion criteria were neoadjuvant therapy, missing data on ≥5 outcome variables and patients were excluded if it was not clearly stated that a prior local resection did not take place.

### 2.2. Clinical Variables

Demographic patient characteristics (sex, age, comorbidity, body mass index) and clinical data (staging MRI, procedure-related parameters, histology parameters, adverse events, 90-day mortality) were collected. Adverse events were subdivided into surgical (anastomotic leak, abscess, bleeding, ileus, fascial dehiscence, perforation, urethral or bladder injury, surgical site infection) and non-surgical (e.g., pulmonary, cardiac, thrombotic, infectious, neurological). If applicable, multiple adverse events were recorded for one patient. Furthermore, data regarding reinterventions and outcomes of sustained injuries, as a result of an adverse event, were collected. For the CS group, additional clinical data was collected (ESD procedure-related parameters, tumor morphology, additional histology parameters, indication for completion surgery).

Surgical resections (primary and completion) were grouped according to anatomic location. Surgical segmental resections of the sigmoid were analyzed as sigmoid resections and all surgical segmental resections of the rectum (e.g., low anterior resection, abdominoperineal resection) were analyzed as Total Mesorectal Excision (TME).

Retrospectively analyzed ESD procedures in the completion surgery group were performed at the discretion of an experienced endoscopist (AK, WG, PD, LM, JH, JB). An *en bloc* resection was defined as macroscopic removal of the lesion in a single piece. Reasons for possible early termination of the ESD without complete lesion removal or conversion to a different resection technique were recorded.

### 2.3. Histology

Tumor stages were histologically confirmed and defined according to the TNM classification: pT1 as invasion through the muscularis mucosae and into, but not beyond the submucosa and pT2 as invasion into the muscularis propria [16]. In case ESD resection was incomplete but completion surgery showed no residual cancer in the surgical specimen, the cancer was staged as pT1. Tumor radicality was subdivided into positive resection margins (R1), unsure radicality (Rx) and radical resection (R0) defined as cancer-free deep and lateral resection margins at histology.

### 2.4. Statistical Analysis

Data were analyzed using SPSS 24 (SPSS, Chicago, IL, USA). Categorical data are expressed as frequencies and percentages. Continuous data are expressed as mean with standard deviation, when normally distributed, and as median with interquartile range (IQR) if data was not distributed normally. The Pearson χ^2^ was used to compare categorical data. Continuous variables were compared using the Mann–Whitney U test. Morbidity outcomes and the amount of harvested lymph nodes were compared between the groups, using logistic regression. In our analysis the amount of harvested lymph nodes was dichotomized with a cut-off of 12. Differences in baseline characteristics were addressed by the use of propensity adjustment; which corrects for these differences without an undesirable reduction in sample size [17]. To estimate the propensity score, logistic regression was performed with the following variables: age, sex, body mass index, American Society of Anesthesiologists (ASA) score, CRC location (sigmoid, rectosigmoid or rectum), modality of procedure (laparotomy, laparoscopy, transanally or robot-assisted) and type of surgery (TME or sigmoid resection). Missing data on propensity score variables or outcome variables were imputed, under the assumption that data were missing (completely) at random. A total of 10 datasets were imputed. A *p*-value ≤ 0.05 was considered statistically significant. A subgroup analysis was performed in the subgroup of TMEs.

## 3. Results

### 3.1. Patient Characteristics

A total of 411 patients were included in the study; 54 patients in the CS group and 357 patients in the PS group. A flow diagram of the patient selection process is shown in Figure 1. Baseline characteristics of both groups, prior to propensity score adjustment, are shown in Table 1. Demographic characteristics did not differ significantly between both groups. ASA-score and tumor location did differ significantly. Patients in the CS group mainly underwent a TME (37/54, 68.5%) whilst patients in the PS group were mainly treated by a sigmoidal resection (221/357, 61.9%). CS and PS were mostly performed laparoscopically (81.5% and 76.8%, respectively).

### 3.2. ESD Characteristics

ESDs in the CS group were *en bloc* in 35/54 patients (64.8%). In 13 patients (24.1%), the ESD was terminated without complete tumor removal due to the suspicion of deep submucosal invasion during the procedure. In six patients (11.1%), the ESD procedure was converted to a piecemeal endoscopic mucosal resection (pEMR). Conversion to pEMR was decided as a result of difficult endoscopic access in one patient and in five patients due to the presence of fibrosis. Five ESDs were complicated by a microperforation or macroperforation. All were managed endoscopically, either by hemoclips (four patients) or an over-the-scope clip (one patient). At histological assessment, 39 cancers were staged as pT1 (72.2%) and 15 as pT2 (27.8%). Of the 35 patients with an *en bloc* resection, 15 patients had a R0 resection (42.9%). Additional clinical data of the completion surgery group are shown in Table 2.

### 3.3. Outcomes of Completion Surgery

The indication for completion surgery was an incomplete resection in 39 patients (72.2%) and the presence of high-risk features in 15 (27.8%). Details on the high-risk features of the pT1CRC subgroup are shown in Appendix A. All patients underwent a radical (R0) oncological resection. In the surgical specimen, a local endoluminal cancer rest was found in 19/39 (48.7%) non-radical (Rx/R1) ESD resections. More information regarding local endoluminal cancer rests is shown in Figure 2. On average, 15.5 lymph nodes were harvested (SD = 10.0). The stoma rate was 20.4% (11 patients), of which three were temporary. Adverse events occurred in 13 patients (24.1%). In nine patients these were classified as surgical adverse events. Anastomotic leak was the most common, occurring in six patients (11.1%). All surgical adverse events required a reintervention. No patients died within 90 days after completion surgery. More outcome variables of the CS group are shown in Table 3.

### 3.4. Outcomes of Primary Surgery

An overview of the outcome variables of the PS group is shown in Table 3**.** The average number of harvested lymph nodes in the PS group was 15.7 (SD = 7.9). The stoma rate was 12.0% (43 patients), of which 24 were temporary. Adverse events occurred in 76 patients (21.3%). In 50 these were classified as surgical adverse events. Anastomotic leak was the most common, occurring in 19 patients (5.3%). Within 90 days after primary surgery one patient died (0.3%).

### 3.5. Comparison between Completion Surgery and Primary Surgery

Outcomes of the comparison between the CS group and PS group after propensity score adjustment are shown in Table 4. No statistical difference was observed in the number of patients in which ≥12 lymph nodes were harvested (OR 0.687; 95%-CI 0.365–1.293, *p* = 0.245). Additionally, no statistical difference was observed in stoma rate (OR 0.864 95%-CI 0.298–2.502, *p* = 0.787), the overall adverse event rate (OR 1.192; 95%-CI 0.514–2.763, *p* = 0.682) or the occurrence of surgical adverse events (OR 1.343; 95%-CI 0.527–3.422, *p* = 0.537). Additionally, no statistical difference was observed for the other morbidity-related outcomes. The 90-day mortality could not be compared because no events occurred in the completion group.

### 3.6. Rectal Surgery Subgroup

In total, 37 patients from the CS group and 136 patients from the PS-group underwent a TME. The average number of harvested lymph nodes in the CS group and PS-group were 16.6 (SD = 11.7) and 16.3 (SD = 8.8), respectively. Stoma rate was 24.3% and 22.8%, respectively. Adverse events occurred in 29.7% and 27.2%, respectively. In both groups, anastomotic leak was the most common adverse event, which occurred in 6 (16.2%) and 12 (8.8%) patients, respectively. The 90-day mortality of both groups was zero. After propensity score adjustment, no statistical association was found between the type of resection (completion or primary) and the number of harvested lymph nodes and morbidity-related outcomes (Appendix A). 

## 4. Discussion

This study evaluated the influence of prior ESD on the surgical morbidity and mortality of completion surgery in suspected T1CRCs in a Western setting. After propensity score adjustments for differences at baseline, no significant difference was seen in 90-day surgical morbidity or survival between the Completion Surgery (CS) group and the Primary Surgery (PS) group. Our findings suggest that ESD does not compromise morbidity and 90-day mortality of completion surgery. 

Previous studies related to this topic have only reported outcomes of CS after ESD without comparing outcomes to a PS group [18], or reported outcomes of completion surgery after endoscopic resections in general [19]. Compared to the only other study reporting results of a cohort of patients undergoing CS following ESD, our overall adverse event rate of 24.1% in the CS-group appears to be slightly higher than the reported 17% [18]. However, the proportion of patients with rectal cancers, associated with a higher risk of adverse events [20], was considerably higher in our study (68.5% vs. 37.7%). This study did not compare their CS group to a PS group and therefore did not answer the question of whether a prior ESD increases the morbidity of completion surgery. 

ESD is a complex endoscopic resection technique with generally longer procedure times and a higher chance of coagulation-induced deep thermal injury, compared to more conventional snare-based endoscopic resection techniques [21,22]. Therefore, the influence on completion surgery for ESD should be studied separately from other snare-based techniques and ESD might show more resemblance to local surgical resection techniques, such as TEM or TAMIS. The influence of these local surgical resections on the morbidity of completion surgery has been studied more frequently. A recent meta-analysis on this subject reported a significant increase in adverse events that required a reintervention if a surgical resection was preceded by a local surgical resection [12]. This is in contrast to our results, where no increase in adverse event rate was observed between the CS group and PS group. The reported overall adverse event rate of 37.4% after completion TME preceded by local surgical resection was also higher than the 29.7% (TME subgroup only) found in our study. These differences might be explained by the difference in the dissection plane. In contrast to the full-thickness or inter-muscular local surgical resections, the submucosal dissection plane of ESDs is more superficial. Any possible inflammatory response that might cause fibrosis and adhesions might be less extensive after ESD, due to the more superficial dissection plane of the submucosa. 

In contrast, our results are in line with the previously mentioned meta-analysis that compared the morbidity of completion surgery after prior endoscopic resection to primary surgery [19]. This study showed that prior endoscopic resections did not appear to increase surgical morbidity. However, as mentioned before, this study did not focus on ESD and mainly studied the influence of more conventional snare-based endoscopic resections. This study reports the largest cohort of patients undergoing completion surgery after ESD with a comparison to an adjusted primary surgery group. 

The quality of completion surgery also does not appear to be negatively affected by ESD. Firstly, the lymph node yield did not differ between the groups. Using the previously reported quality indicator of 12 lymph nodes as a cut-off point, our study found no significant difference in lymph node yield between the CS-group and the PS-group [23]. In contrast to our results, local surgical resections did appear to significantly reduce the number of harvested lymph nodes [12]. Secondly, prior local surgery was associated with an increase in incomplete mesorectal excisions, using previously described grading of the mesorectum [24]. Although reporting and comparing mesorectal grading was not possible in our dataset due to missing data, we did observe that all completion surgeries were radical (R0) resections. This suggests that the quality of completion surgery may also be unaffected by prior ESD. 

This study has some limitations. Firstly, due to the rarity of this clinical situation, the number of patients in the completion surgery group is limited. This should be taken into account when interpreting the results of the comparison. Secondly, due to our study’s retrospective nature, there is an inherent risk of selection bias, which we have tried to minimize by using propensity score adjustment. Nevertheless, residual confounding cannot be excluded entirely, especially because some relevant characteristics were unavailable, such as the exact tumor location within the rectum, which might be related to surgical complexity [25]. Additionally, since the data used for analysis were collected from different centers, possible variations in the quality of surgical procedures and differences in medical personnel across centers may introduce a form of selection bias. Thirdly, the DCRA is a self-reported surgical database, which brings a risk of under-registration. This database was also not specifically designed for this study, which is why some relevant information, for example on histological high-risk features, was not available for the primary surgery group. In addition, we had to exclude 2876 patients, because it was not clearly stated if a local resection took place. Although missingness was unlikely to be related to the outcome, we were not able to completely exclude the possibility of some selection bias. Fourthly, ESDs were included starting from 2014, when the procedure was still being introduced in the West. As a result, ESD performance might be slightly inferior to performance in current practice. However, by selecting only ESDs after which completion surgery was performed and thus, excluding all definitive ESD procedures, the quality of included ESDs is not representative for the general ESD performance in these centers. Lastly, due to unavailable data, we were unable to study the influence of a prior ESD on functional outcomes or procedure times of completion surgery. 

This study has implications for clinical care as it adds to the previous evidence that it is safe to perform ESDs for suspected T1CRCs without compromising completion surgery. The long-term oncological safety of this strategy was previously reported [7,8,11]. Additionally, it was previously reported that the perceived time to recovery after completion surgery does not appear to differ from primary surgery [26]. Our current study shows that ESDs for suspected T1CRCS do not appear to increase surgical morbidity and 90-day mortality. Taken together it seems justified to perform ESD in all patients with a suspected T1CRC, to prevent extensive surgery for the substantial number of patients with a low-risk T1CRC. To increase the validity of the present study, performing a follow-up clinical trial may be considered.

## 5. Conclusions

This study suggests that ESD does not adversely affect the morbidity and 90-day mortality of completion surgery.

## Figures and Tables

**Figure 1 cancers-15-04490-f001:**
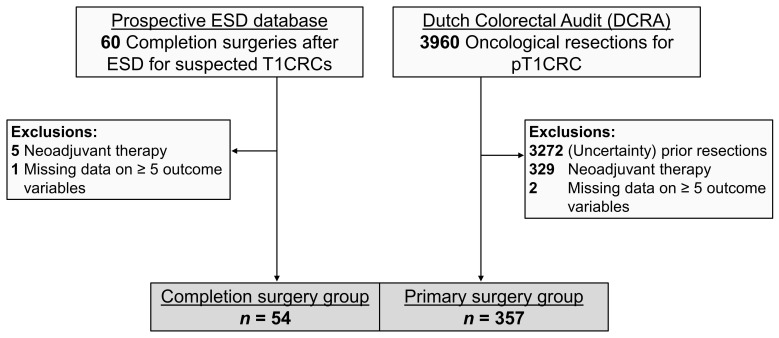
Flow diagram of patient selection. ESD, endoscopic submucosal dissection; CRC, colorectal cancer.

**Figure 2 cancers-15-04490-f002:**
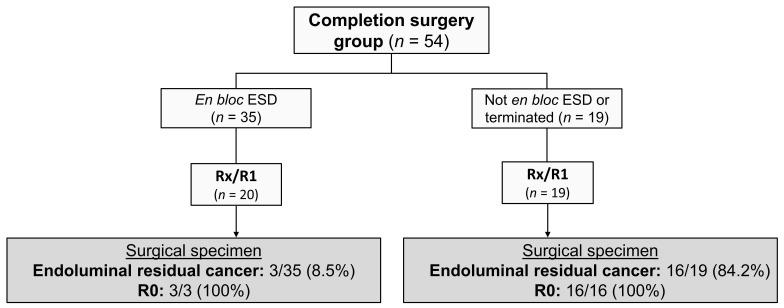
Outcomes of the completion surgery group. *ESD*, endoscopic submucosal dissection. *R0*, radical resection; *Rx*, unsure radicality; *R1*, non-radical.

**Table 1 cancers-15-04490-t001:** Baseline characteristics of study participants prior to propensity score adjustment.

	Completion Surgery (*n* = 54)	Primary Surgery (*n* = 357)	*p* Value
Sex, male	35 (64.8)	209 (58.5)	0.382
Age, years, mean (SD)	66.9 (8.63)	67.1 (9.16)	0.258
BMI, kg/m^2^, mean (SD)	28.4 (5.67) Unknown (*n* = 9) ^1^	27.6 (4.61) Unknown (*n* = 10) ^1^	0.169
ASA-score I II III IV	10 (18.5) 40 (74.1) 4 (7.4) 0 (0)	63 (17.6) 207 (58.0) 86 (24.1) 1 (0.3)	0.043
Tumor location Sigmoid Rectosigmoid Rectum	15 (27.8) 14 (25.9) 25 (46.3)	213 (59.7) 35 (9.8) 109 (30.5)	<0.001
Type of surgery Sigmoid resection TME	17 (31.5) 37 (68.5)	221 (61.9) 136 (38.1)	<0.001
Surgical approach Laparoscopic Open taTME Robot-assisted	44 (81.5) 3 (5.6) 2 (3.7) 5 (9.3)	274 (76.8) 10 (3.1) 11 (3.1) 46 (12.9) Unknown (*n* = 16) ^1^	0.646

^1^ Prior to analyses, missing data were imputed, using multiple imputations with 10 iterations. SD, standard deviation; BMI, body mass index; ASA, American society of anesthesiology; taTME, trans-anal total mesorectal excision.

**Table 2 cancers-15-04490-t002:** Clinical data of the completion surgery group.

	**Completion surgery group** **(*n* = 54)**
**Tumor characteristics**	
Diameter polyp, mm (IQR)	25.0 (22.5). Unknown (*n* = 4)
Gross morphology Sessile Flat Pedunculated	35 (64.8) 12 (22.3) 4 (7.4) Unknown (*n* = 3)
Staging MRI performed prior to ESD	10 (18.5) ^1^
**Technical details ESD**	
Duration, median (IQR)	129 min (103) Unknown (*n* = 11)
Perforation (microperforation or perforation)	5 (9.3)
*En bloc* Yes No	35 (64.8) 19 (35.2)
Radicality R0 R1/Rx	15 (27.8) 39 (72.2)
Tumor stage ESD specimen pT1 pT2	39 (72.2) 15 (27.8)
Subsequent eFTR performed	3 (5.6)
**Completion surgery**	
Indication additional therapy ^2^ Not R0 resection High-risk histology	39 (72.2) 15 (27.8)
Time to surgery, days, median (IQR)	56.5 (37)

^1^ For patients eventually undergoing TME this was 10/37 (27.0%). Specific anatomic location within the rectum was reported in 6 patients. ^2^ In case a surgical specimen showed both Rx/R1 and an additional high-risk criterium, the indication was scored as *not R0 resection*. Values are *n* (%) unless otherwise defined. ESD, endoscopic submucosal dissection; IQR, interquartile range; MRI, magnetic resonance imaging; pEMR, piecemeal endoscopic mucosal resection; eFTR, endoscopic full-thickness resection.

**Table 3 cancers-15-04490-t003:** Detailed outcome variables before propensity score adjustment for completion surgery and the primary surgery groups.

	Completion Surgery (*n* = 54)	Primary Surgery (*n* = 357)
Stoma after resection Temporary ileostomy Permanent ileostomy Temporary colostomy Permanent colostomy	11 (20.4) 3 0 0 8	43 (12.0) 23 1 1 18
Adverse events < 90 days	13 (24.1)	76 (21.3)
Surgical adverse events Anastomotic leak Abscess Bleeding Ileus Fascial dehiscence Perforation Urethral or bladder injury Surgical site infection	9 (16.7) 6 0 0 2 0 0 2 0	50 (14) 19 8 1 11 0 2 1 8
Non-surgical adverse events Pulmonary Cardiac Thrombotic Infectious Neurological Other	6 (11.1) 1 1 0 1 2 4	42 (11.8) 12 5 3 10 0 18
Reintervention ^1^ Laparotomy Laparoscopy Endoscopy Radiology Other	9 (16.7) 2 5 1 0 1	31 (8.7) 10 11 2 1 4 Unknown (*n* = 3)
Stoma by reintervention ^1^ Temporary ileostomy Permanent ileostomy Temporary colostomy Permanent colostomy Unknown	6 (11.1) 4 1 1 0 0	18 (5.0) 7 0 4 7 3
ICU admission ^2^ Median stay, days (range)	1 (1.9) 4	17 (4.8) Unknown *n* = 4 2 (1–12)
Permanent injury	2 (3.7)	1 (0.3)
Lymph nodes harvested, mean (SD)	15.5 (10.0)	15.66 (7.9)
90-day mortality	0	1 (0.3)

^1^ As result of a surgical adverse event. ^2^ As result of an adverse event. Values are *n* (%) unless otherwise defined. ICU, intensive care unit; SD, standard deviation.

**Table 4 cancers-15-04490-t004:** Comparison between completion surgery and primary surgery groups.

Outcome Variable	Odds Ratio (95% CI)	*p* Value
Lymph nodes harvested ^1^	0.687 (0.365–1.293)	0.245
Stoma after surgery	1.298 (0.587–2.872)	0.519
Adverse event < 90 days Surgical adverse event Reintervention required Stoma by reintervention ICU admission as result of an adverse event Permanent injury	1.162 (0.570–2.370) 1.133 (0.498–2.576) 1.572 (0.661–3.737) 1.864 (0.651–5.335) 0.210 (0.025–1.737) 2.937 (0.246–35.115)	0.679 0.767 0.306 0.246 0.148 0.391
90-day mortality	NA	NA

^1^ Variable was dichotomized, using 12 lymph nodes as cut-off. The primary surgery group was used as reference for the regression analysis. CI, confidence interval; ICU, intensive care unit.

## Data Availability

The data that support the findings of this study are available from DCRA but restrictions apply to the availability of these data, which were used under license for the current study, and so are not publicly available. Data are however available from the authors upon reasonable request and with permission of DCRA (contact for data request is corresponding author N. Dekkers).

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
