# Peer review of "Outcome of Completion Surgery after Endoscopic Submucosal Dissection in Early-Stage Colorectal Cancer Patients"

_cancers, 2023, doi:10.3390/cancers15184490_

Round 1
Reviewer 1 Report
This study is very interesting and it has a very important clinical impact.
This is an interesting and well conducted retrospective study that demonstrates that ESD (Endoscopic Submucosal Dissection) does not adversely affect the morbidity and 90-day mortality of completion surgery by comparing the morbidity of completion surgery after ESD to primary surgery in a Western population of Early Stage Colorectal Cancer Patients.
The manuscript is clear, comprehensive; the topic is original and has implications for clinical care because the influence of ESD on surgical morbidity had never been analyzed before.
The introduction provides sufficient background and include all relevant references. The research design is appropriate, and the methods are adequately described.
The results are clearly presented and provide an advancement of the current knowledge with conclusions supported by the results.
Tables and figures are clear and represent data in a straightforward manner.
English language fine.
Author Response
Dear reviewer 1,
Thank you very much for taking the time to review this manuscript and the kind words regarding the manuscript. Please find the detailed responses below and the corresponding revisions/corrections highlighted in the re-submitted files.
Comments and suggestions
There were no comments or suggestions requesting editing.
English language
Based on the 'English language is fine' comment, we have carefully re-read the manuscript with a native English speaker and made a few adjustments to increase readability and English language of the manuscript:
- Row 21: "find out" was replaced by "investigate" instead of "find out".
- Row 22, 23: "that" was replaced by "who"
- Row 24: "was" was corrected to "were"
- Row 41: "analyses" was corrected to "analysis"
- Row 65: "has" was corrected to "have"
- Row 72-73: "whether the same applies for" was corrected to "of whether the same applies to"
- Row 110: "a" and a comma were added
- Row 128: a hyphen was added
- Row 167: "a" was added and a typo was corrected
- Row 176: "the" was added
- Row 180: "are" was corrected to "is"
- Row 181: "the" was added
- Row 245 & 246: "the" was added
- Row 259: "the" was added
- Row 267: "the" was added
- Row 315: "the" was added
- Row 318: "a" was added
- Row 328: "the" was added
- Row 332: "Current" was replaced by "this"
- Row 248: a comma was added and "for" was corrected by "of"
We appreciate your assistance in enhancing the quality of our work. If you have any further comments or concerns, please do not hesitate to let us know. Your input is greatly appreciated.
Kind regards,
Also on behalf of the coauthors,
Nik Dekkers
Reviewer 2 Report
This is a nicely written manuscript of a study that compared the outcome of completion surgery (CS) after endoscopic submucosal dissection (ESD) in early stage colorectal cancer and primary surgery (PS) for colorectal cancer. The study results show that post op surgical morbidity and 90 day mortality rates were similar for both procedures. There were no statistically significant differences in the odds of morbidity and 90 day mortality between the two procedures.
A few suggestions are listed below in order to further improve the quality of the paper.
1. Simple summary: In line 20, use measurable terms like "to investigate" (Bloom's taxonomy) instead of "to figure out". In line 24, revise the term "ESD does not impact the safety and outcome of surgery". The reality is that ESD does actually impact the safety and outcome of surgery, though the differences in negative outcomes between ESD+CS and PS is not significant.
2. Abstract: The sentence in line 35 "A subgroup analyses was performed in patients undergoing rectal surgery" should be moved to line 41 just before the statement of results for the rectal surgery group. Include the total number of study participants (N=411) in the abstract.
3. Introduction: In line 64, revise the statement "The influence of local resections on the morbidity of completion surgery has mostly been studied for prior local resections ..." In reality, local resection is not done with the purpose of having morbidity in patients. You may consider "Morbidity rates of completion surgery following local resections has mostly been studied..."
4. Materials and Methods: In line 106, give some examples of the non-surgical adverse events. In line 115-118, the statement here does not suggest that a retrospective cohort study was done. Unless the cancer registry recorded that the procedures were performed by the listed individuals. This should be clearly explained here. In line 129, are "absolute numbers" the same as "frequencies"? If so, I suggest you use the term "frequencies".
3. Results: The results are well described. However, the tables should be revised. In table one the title should be "Baseline characteristics of study participants prior to propensity score adjustment". All other additions to the title should come under the footnote. The same for table 2 and figure 2. Table 3 should read "Detailed outcome variables before propensity score adjustment for completion surgery and the primary surgery groups. Move the added sentence to the footnote. Table 4, Revise the title to "Comparison between completion surgery and primary surgery groups." Move the added sentence to the footnote. Also, for table 4, indicate which surgery group was used as the reference for the regression analysis.
Discussion: This is nicely presented. I suggest that you also state in the limitations that since the data used for analysis were collected from different centers, there may be variations in the quality of the surgical procedures and between the medical personnel across centers therefore presenting a form of selection bias. You may recommend a follow-up clinical trial or randomized control trial in order to increase the validity of the present study.
I have attached a file with highlighted texts for your convenience during the revision process.
Best of luck!
